# Boronium Salt as an Antiviral Agent against Enveloped Viruses Influenza A and SARS-CoV-2

Terrence J. Ravine [1],*, Jonathan O. Rayner [2], Rosemary W. Roberts [2], James H. Davis, Jr. [3] and Mohammad Soltani [3]

1. Department of Biomedical Sciences, University of South Alabama, Mobile, AL 36688, USA
2. Department of Microbiology & Immunology, University of South Alabama, Mobile, AL 36688, USA
3. Department of Chemistry, University of South Alabama, Mobile, AL 36688, USA
* Correspondence: travine@southalabama.edu

**Abstract:** Quaternary ammonium compounds (QACs) are routinely used as disinfectants in a variety of settings. They are generally effective against a wide range of microbes but often exhibit undesirable toxicity. Consequently, companies are constantly seeking alternatives to QACs that are just as effective but with reduced health and environmental hazards. Two boronium salt derivatives were tested against influenza A and SARS-CoV-2 viruses. One salt possessed a terminal benzyl group, while the other lacked the same terminal benzyl group. Both salts demonstrated virus inactivation similar to a commercial QAC disinfectant. The non-benzylated form exhibited the same cell toxicity profile as the QAC. However, the benzylated form displayed less cell toxicity than both the non-benzylated form and QAC. These results suggest that the boronium salts may be suitable for use as a disinfecting agent against enveloped viruses in lieu of using a QAC. Continued evaluation of the boronium salts is warranted to determine the lowest effective concentration capable of effectively controlling influenza A and SARS-CoV-2 viruses that also demonstrates low cytotoxicity.

**Keywords:** boronium ion; disinfectants; quaternary ammonium compound; cytotoxicity; antiviral

## 1. Introduction

Quaternary ammonium compounds ($R_4N^+$), also known simply as quats or QACs, are effective for low-level disinfection of many surfaces. Most quats are cationic compounds containing the organic salt benzalkonium chloride. QACs are generally soluble in water and alcohol as long as their chain length does not exceed C14 [1]. They act by disrupting cell membranes, allowing intracellular contents to exit the cell. They are mainly bacteriostatic when used at low concentrations and are bactericidal at higher concentrations. Commercial QACs are usually seen as a mixture of compounds with differing alkyl chain lengths, ranging from C8 to C18. Greater biocide activity is seen with C12 and C14 lengths [2].

QACs are widely used as antiseptics, disinfectants, preservatives, and sterilization agents in a variety of settings, such as homes, medical facilities, and water treatment facilities. They are also extensively used in both the textile and food industries. QACs are also employed as herbicides and pesticides, although environmental safety concerns are associated with their use. For instance, the popular QAC herbicide paraquat dichloride (paraquat) has been banned for use in several countries, including the United States, due to toxicity [1].

Toxicity and biosafety profiles are both top priorities for chemical agents with possible human contact that will also eventually make their way into the environment. In this regard, the safety profile for boronium salts has yet to be established. However, it is well known that QACs such as those found in Bacdown disinfectant pose both health and environmental hazards. The Agency for Toxic Substances and Disease Registry (ATSDR) states that Category 1 agents pose an "Urgent public health hazard", while Category 2 agents carry a "Public health hazard" [3]. The Safety Data Sheet (SDS) [4] indicates the following health and environmental hazard levels:

- Skin corrosion/irritation (Category 2)
- Serious eye damage/eye irritation (Category 1)
- Hazardous to the aquatic environment, acute hazard (Category 1)
- Hazardous to the aquatic environment, long-term hazard (Category 2)

Moreover, increased global production and use of synthetic QACs carries with it similar environmental and health hazards [1]. Most notably:

- General toxicity
- Accumulation of QACs in soil, sludge/sewage, water, and plants
- Bacterial and antibiotic resistance and co-/cross-resistance
- Risk of asthma

While QACs are effective against several bacterial types, some bacteria, such as pseudomonads, are not adequately controlled by these compounds. In fact, some *Pseudomonas* strains are known to grow (multiply) in QACs. Furthermore, viruses, bacterial spores, and some fungi are recognized to be more resistant to the action of QACs and require a different type of intermediate-level disinfection [5].

Two boronium salts with neutral charges were synthesized, each containing a boronium ion attached to a C16 alkyl tail, as previously characterized [6]. One salt possessed a terminal benzyl group (B) while the other salt lacked a benzyl group (A), as is seen in Figure 1. Both salts were subsequently tested against three different bacteria. The benzylated form was demonstrated to be four times more effective than the commercial QAC Bacdown disinfectant detergent (D + E), as well as BDD (Decon Labs, King Of Prussia, PA), at inhibiting the growth of both methicillin-sensitive *Staphylococcus aureus* (MSSA) and *Escherichia coli*. Conversely, the non-benzylated form did not display growth inhibition in any test bacterium at the highest concentration tested. The most noteworthy finding was that the benzylated form was eight times more effective against *Pseudomonas aeruginosa* than the Bacdown QAC [6].

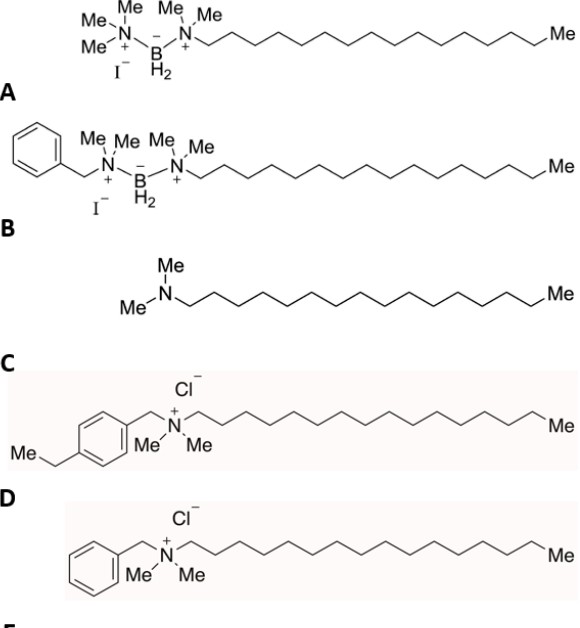

**Figure 1.** Test articles. Chemical structures of (**A**)—boronium salt without benzyl group ($C_{21}H_{50}BIN_2$), (**B**)—boronium salt with benzyl group ($C_{27}H_{54}BIN_2$), (**C**)—N,N-dimethylhexadecylamine, a negative control without boronium ion ($C_{18}H_{39}N$), (**D**)—alkyl dimethyl ethylbenzyl ammonium chloride ($C_{27}H_{50}ClN$), (**E**)—alkyl dimethyl benzyl ammonium chloride ($C_{25}H_{46}ClN$). Structures D and E are similar to those found in the Bacdown disinfectant, which acted as a positive control (C16 structures are shown).

Subsequently, these same two boronium salts were tested against two opportunistic fungal pathogens, namely *Candida albicans* yeast and *Aspergillus fumigatus* mold. Yeasts grows at higher temperatures closer to the human body temperature of 37 °C, whereas molds tend to favor a lower, more ambient (room) temperature of 25 °C [7]. Here, the non-benzylated form showed better control of *C. albicans* yeast growth and equal control of *A. fumigatus* mold growth when compared to the Bacdown QAC disinfectant. In contrast, the benzylated form was less effective than either the non-benzylated form or Bacdown QAC against either fungus type [8].

Differences seen in the boronium salts' antibacterial and antifungal activity are most likely attributed to variations in bacterial and fungal membranes. The underlying plasma membrane is the target region, where the bornium salts are thought to have a disruptive effect. Bacteria are classified as prokaryotes, while fungi are eukaryotes. One such membrane difference is sterols, like ergosterol present in fungal membranes, which are absent in prokaryotes. Instead, bacterial membranes contain hopanoids, a sterol-like lipid [6–8].

During maturation and prior to leaving the cell, enveloped viruses acquire their envelope from host cell membrane, such as the plasma membrane or other intracellular membranes, depending on the virus. Disruption of the outer virus envelope has been demonstrated by QACs [9]. Plasma membrane disruption is also the suspected mechanism for boronium salts' activity against bacteria and fungi [6,8]. This suggested an additional assessment of boronium salts for their ability to similarly inactivate infectivity of enveloped RNA viruses, as a potential alternative to QACs.

To accomplish this, the first step was to determine if either boronium salt exhibited any antiviral activity against two notable enveloped RNA viral pathogens, namely influenza A and SARS-CoV-2. Serial dilutions of each test article had to be prepared, but it was unknown what a reasonable starting concentration should be that would not negatively impact the health of culture cells in permissible host tissue. Results from prior bacterial and fungal studies could not be used as a predictor of what concentration would be needed to show antiviral activity. An additional study facet was to determine what effect each boronium salt had on these tissue culture cells. An estimation of cell toxicity caused by each test article was also of great interest.

The purpose of this investigation was to establish initial concentration breakpoints of each test article to be used during later definitive testing. The authors wanted to see how each boronium salt compared to an approved virucidal QAC disinfectant, such as Bacdown, when adjusted to an equal concentration and not to individual Bacdown components. During this process, both antiviral effectiveness and cell cytotoxicity were examined. We now present the results of preliminary testing using two frequently encountered viral pathogens, namely influenza A (INFA) and SARS-CoV-2.

## 2. Results

### 2.1. Virus Inactivation

The degree to which a specific test compound inactivated INFA and SARS-CoV-2 viruses was evaluated after a 1 h co-incubation, as described in the Section 4. INFA virus titers decreased in comparison to the untreated control virus when incubated with as little as 9.6 µg/mL of Bacdown QAC, the boronium salt with benzyl group, and the aliphatic compound lacking the boronium group (Table 1).

A minimum of 48 µg/mL of the boronium salt without benzyl group was required to decrease INFA titers in comparison to the untreated control. In contrast, the boronium salt w/benzyl group, boronium salt without (w/o) benzyl group, and Bacdown QAC all reduced INFA virus titers below the limit of detection (LOD) at concentrations ≥240 µg/mL, while the aliphatic compound w/o boronium group required concentrations ≥1200 µg/mL to achieve inactivation of INFA virus below the LOD.

For SARS-CoV-2, a concentration > 48 µg/mL of the boronium salt w/benzyl group, boronium salt w/o benzyl group, and Bacdown QAC was required to decrease infectious virus titers in comparison to the untreated control: however, all three demon-

strated virus inactivation below the LOD at concentrations ≥ 240 µg/mL like that seen with INFA (Table 1). These results suggest that while SARS-CoV-2 virus is perhaps more stable than INFA virus when exposed to lower concentrations of these test compounds, the optimal effective concentration for eliminating both viruses is similarly between 48 and 240 µg/mL. In contrast, the aliphatic compound w/o boronium required a concentration > 240 µg/mL to decrease SARS-CoV-2 virus titers, and inactivation below the LOD required concentrations ≥ 6000 µg/mL. Therefore, the aliphatic compound w/o boronium is less effective at inactivating SARS-CoV-2 virus as compared to INFA.

**Table 1.** Results of test article virus inactivation capability.

| Test Article | Concentration at which Virus Titer Fell below the LOD (µg/mL) | |
| --- | --- | --- |
| | INFA | SARS-CoV-2 |
| Quaternary ammonium compound | ≥240 | ≥240 |
| Boronium ion w/o benzyl group | ≥240 | ≥240 |
| Boronium ion w/benzyl group | ≥240 | ≥240 |
| Aliphatic compound w/o boronium ion | ≥1200 | ≥6000 |

*2.2. Cell Toxicity*

Each of the test compounds was expected to be toxic to the cells, potentially confounding the assessment of infectious virus titers in the tissue culture assay. Thus, the degree by which each test compound alone caused cell death at each concentration was assessed in both MDCK and Vero cells. The boronium salt without benzyl group and Bacdown QAC both demonstrated MDCK cell toxicity at concentrations ≥ 240 µg/mL. In contrast, the boronium salt with benzyl group and aliphatic compound lacking the boronium group were less toxic in MDCK cells, requiring concentrations ≥ 1200 µg /mL to cause toxicity (Table 2). Vero cells in general were less subject to the toxic effects of all test compounds when compared to MDCK cells. Specifically, the boronium salt without benzyl group and Bacdown QAC both caused cytotoxicity in Vero cells at concentrations ≥ 1200 µg/mL, while the boronium salt with benzyl group required concentrations ≥ 6000 µg/mL to cause cytotoxicity. Yet, the aliphatic compound lacking the boronium group had no toxic effect on Vero cells at any of the concentrations tested (Table 2).

**Table 2.** Results of test article cell toxicity.

| Test Article | Concentration at which Cell Toxicity Was First Observed (µg/mL) | |
| --- | --- | --- |
| | MDCK | Vero |
| Quaternary ammonium compound | ≥240 | ≥1200 |
| Boronium ion w/o benzyl group | ≥240 | ≥1200 |
| Boronium ion w/benzyl group | ≥1200 | ≥6000 |
| Aliphatic compound w/o boronium ion | ≥1200 | ND |

ND = none detected.

**3. Discussion**

*Overall Assessment*

Based on all assay results, the boronium salt with a terminal benzyl group (B) displayed the best combination of both virus inactivation and decreased cell toxicity when compared to Bacdown QAC (D + E). Both compounds were equally effective at inactivating INFA and SARS-CoV-2 virus, with virus titers falling below the LOD at ≥240 ug/mL. However, the benzylated boronium salt was significantly less toxic to the cells, requiring concentrations ≥ 11200 µg/mL and ≥6000 µg/mL to cause cytotoxicity in MDCK and Vero cells,

respectively. Bacdown QAC caused MDCK cell toxicity at concentrations $\geq 240\ \mu g/mL$ and Vero cell toxicity at concentrations $\geq 1200\ \mu g/mL$ (Table 1). Importantly, the concentration at which MDCK cell toxicity was first observed for Bacdown QAC is the same as the concentration at which INFA titers fell below the LOD, supporting concerns regarding the toxicity of this compound at effective concentrations. In contrast, the boronium salt with benzyl group required greater concentrations to cause toxicity in both MDCK and Vero cells when compared to the concentration required to decrease INFA and SARS-CoV-2 titers below the LOD, suggesting that it would be safer to use as a disinfectant when compared to Bacdown QAC.

The boronium salt without the benzyl group (A) was also equally effective as Bacdown QAC (D + E) at inactivating both INFA and SARS-CoV-2 ($\geq 240\ \mu g/mL$); however, it exhibited the same cell toxicity profile as Bacdown QAC in both MDCK ($\geq 240\ \mu g/mL$) and Vero cells ($\geq 1200\ \mu g/mL$), making it less attractive as an alternative to Bacdown QAC (Table 1). The aliphatic compound without boronium ion (C), in contrast, demonstrated MDCK cell toxicity at concentrations $\geq 1200\ \mu g/mL$ and no toxicity on Vero cells, which is similar to or better than the boronium salt with benzyl group. However, higher concentration of the aliphatic compound without boronium ion was required for INFA ($\geq 1200\ \mu g/mL$) and SARS-CoV-2 ($\geq 6000\ \mu g/mL$) virus inactivation as compared to the boronium salt with benzyl group (B) or Bacdown QAC ($\geq 240\ \mu g/mL$) negating its utility as an effective disinfectant.

With respect to the antiviral mechanism of action, QACs are well characterized as cationic surfactants containing a positively charged/hydrophilic nitrogen head and a long chain hydrophobic tail [1]. They are known to interact with key virus elements of DNA/RNA, proteins, and lipids [9]. QACs are recognized to be effective against hydrophobic enveloped viruses, such as human immunodeficiency virus, hepatitis B, herpes simplex, influenza, and severe acute respiratory syndrome (SARS-CoV). Their effectiveness is a result of interacting primarily with hydrophobic outer viral envelope phospholipids.

Both influenza and SARS-CoV-2 outer membrane phospholipids share similarity with bacterial membranes, with the exception of some variation in bacterial phosphate heads [10]. The QAC benzalkonium chloride interdigitates within bacterial and yeast plasma membranes via its alkyl chains, thereby disrupting cell membrane function [11]. Likewise, QACs are thought to disrupt viral envelopes by this same mechanism. QACs are less effective against hydrophilic nonenveloped viruses, such as adenovirus, enterovirus, rhinovirus, and rotavirus [12].

The amphiphilic character of the boronium salts (A, B) may allow them to act in a similar fashion. They could likewise act to disrupt the outer virus lipid envelope during the initial co-incubation phase prior to exposure of permissible culture cells (Figure 2). In doing so, the hydrophobic 16C alkyl chain of the boronium ion inserts itself between adjacent envelope phospholipids, thereby altering the structural integrity of the viral outer membrane. In theory, disruption of the outer membrane should result in loss of embedded INFA hemagglutinin glycoproteins to adversely affect virus attachment. This would result in decreased virus adherence to host cell membranes, preventing subsequent host cell entry.

Computational modeling had previously revealed a noticeable difference in head group dipole movement between the benzylated and non-benzylated boronium salts. This result suggested that each salt head group would interact differently with negatively-charged regions of microbe membranes [8]. The greater antiviral activity exhibited by the boronium salt with benzyl group may be related to a better ability to interact with virus envelope outer membrane phospholipids than the non-benzylated form. In this regard, the chemical nature of each boronium salt's head group is thought to account for these differences. Here, the benzylated form demonstrates more interaction with negatively-charged membrane regions of tested bacterial and fungal microbes, leading to greater membrane disruption [4,6]. Moreover, it is entirely possible that the terminal-located benzyl group may also be interfering with a different viral component, such as viral RNA and/or proteins, as previously indicated for QACs. Similar testing of rhinovirus is being

considered to establish the boronium salt's effectiveness against a nonenveloped virus, where QACs are less effective. It is interesting to note that the alkyl dimethyl benzyl ammonium chloride (C12-18) in the Bacdown disinfectant also has a terminal benzyl group. The presence of the benzyl group alone did not account for the difference seen in the concentration at which cell toxicity was first observed in MDCK and Vero cells.

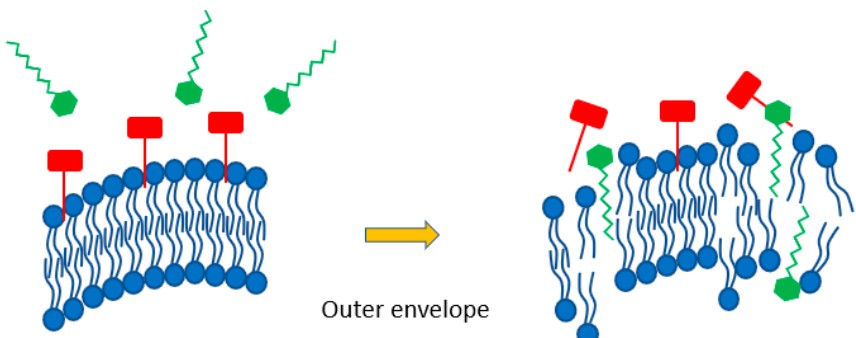

**Figure 2.** Schematic indicating benzylated boronium salt (green) disrupting outer viral envelope phospholipids (blue) causing a release of INFA hemagglutinin glycoproteins (red) that are required for host cell attachment.

Study limitations include testing of a low number of compound concentrations, along with the inherent limit of detection (LOD) in performed virus inactivation and cell cytotoxicity assays. Further testing will be conducted to determine the lowest effective concentration of each test article. This can be accomplished now that targets for effective test article concentrations have been established. Replicate assays assessing virus inactivation are also indicated, although the reported cell toxicity values represent the average of triplicate cultures. This will allow a statistical analysis to determine if differences in effective concentrations are significant. The use of a shorter contact period (e.g., 10–30 min) will also be examined.

The suggestion that the terminal benzyl group bound to one boronium salt (B) exerts an additional antiviral effect is purely speculative. Further testing is required to establish the exact nature of its interaction with non-enveloped virus components or DNA-containing viruses. In this regard, an examination of virus and host cell membrane integrity, pre- and post-agent exposure, may be helpful in revealing any additional virus inactivation mechanism(s). This could be accomplished using a combination of specialized techniques, such as cryo-electron microscopy of vitreous sections (CEMOVIS), electron tomography (ET), and lipid analysis by mass spectrometry [13,14].

Of particular interest was that both boronium salts (A, B) demonstrated equal or reduced cell toxicity to Bacdown QAC (D + E) in Vero cells of primate origin (African green monkey kidney) and MDKC cells of canine origin (normal kidney epithelium). These findings suggest a comparable or decreased cytotoxicity potential of the benzylated salt in human cell lines when compared to Bacdown QAC. Validation of reduced cell toxicity could be accomplished using an MTT assay. The MTT (3-[4,5-dimethylthiazol-2-yl]-2,5 diphenyl tetrazolium bromide) assay is routinely used to assess cell metabolic activity, viability, membrane integrity, and proliferation, all of which are indicators of overall cell health [15]. Here, several different human cell lines could be tested to determine the effects of either boronium salt on both cell viability and cell proliferation.

## 4. Materials and Methods

### 4.1. Cells and Viruses

Madin-Darby canine kidney cells (MDCK cells) [London Line, FR-58] were obtained from International Reagent Resource (Manassas, VA, USA). After in-house adaptation, cells were maintained at 37 °C with 5% $CO_2$ in serum-free media [OptiPRO SFM (Gibco, Waltham,

MA, USA) supplemented with 2X GlutaMAX (Gibco) and 1% Penicillin/Streptomycin (1% Pen/Strep) (100 units/mL Penicillin and 100 µg/mL Streptomycin, Gibco)].

Shell Vial Vero E6 Cells (SV-VeroE6 cells) [Cercopithecus aethiops kidney clone from Vero 76] (Quidel, San Diego, CA, USA) were recovered from glass coverslips and maintained at 37 °C with 5% $CO_2$ in complete media [Dulbecco's Modified Eagle Media (DMEM) (Lonza, Basel, Switzerland) supplemented with 10% Fetal Bovine Serum (FBS) (Millipore, Burlington, Massachusetts), 1% L-glutamine (2 mM, Lonza), and 1% Pen/Strep (100 units/mL Penicillin and 100 µg/mL Streptomycin, Lonza)].

Influenza virus strain A/PR8/34 (Identifier H1N1, Lot # 111519JHK, $4.1 \times 10^8$ Focus Forming Units (FFU)/mL) was rescued using reverse genetics methods by transfection with a 12-plasmid system of influenza A (IAV) A/PR8/34 (H1N1) strain in 293T cells [ATCC, CRL-11268], as described previously [16]. The stock of IAV was prepared by infecting rescued IAV to confluent MDCK cells in T75 flasks for 1 h at 37 °C. The inoculum was removed and MDCK cells were washed twice with Dulbecco's Phosphate Buffered Saline (DPBS) (Gibco). Fresh growth medium [(OptiPRO SFM supplemented with 2% GlutaMAX-1 and 0.5 µg/mL trypsin treated with N-tosyl-$_L$-phenylalanine chloromethyl ketone (TPCK-trypsin)] was added, and the infected MDCK cells were incubated at 37 °C for 3 days. Harvested IAVs were filtered with a 0.45 µm polyethersulfone filter (Fisherbrand, Waltham, MA, USA), divided into aliquots, and stored at −80 °C. The stock of IAV was sequenced after PCR amplification of all 8 segments to ensure that there were no unwanted mutations (GENEWIZ, South Plainfield, New Jersey).

SARS-CoV-2 strain MOBAL-05 (Identifier MOBAL-05, Lot # 110420JOR, $2.92 \times 10^6$ Plaque Forming Units (PFU)/mL) was isolated from a clinical sample in Mobile, Alabama. Emergency Department (ED) swabs in viral media were received from University Hospital (Mobile, Alabama) and were vortexed for 30 s then allowed to settle prior to aliquoting into 1.7 mL cryovials for storage at −80 °C. Samples were thawed and 0.5 mL was inoculated per well onto approximately 80% confluent SV-VeroE6 cells in 12-well tissue culture plates (Corning, Corning, NY, USA). Plates were centrifuged at $1900 \times g$ for 1 h at room temperature to promote interaction between the virus and cells. Plates were incubated at 37 °C with 5% $CO_2$ and monitored daily for Cytopathic Effects (CPE). After 72 h, 0.2 mL of culture supernatant was transferred to a well of ~80% confluent SV-VeroE6 cells in a 6-well plate. The remaining media in the 12-well plate was removed and replaced with 1 mL maintenance media (DMEM supplemented with 2% FBS, 1% L-glutamine, and 1% Pen/Strep); a 1 mL pipette tip was used to scrape cells, pipetting to mix, and 0.2 mL was added to the corresponding supernatant in the 6-well plate. Plates were incubated at 37 °C with 5% $CO_2$ and monitored for CPE. After 72 h culture, supernatants were filtered with a 0.45 µm PVDF syringe filter and aliquoted into cryovials for storage at −80 °C.

Master and Working stocks were confirmed to be free from mycoplasma contamination and had endotoxin levels below 0.5 endotoxin units (EU)/mL. Influenza virus titers were determined via focus forming assay in MDCK cells. At 1 h after inoculation at 37 °C, infected MDCK cells were overlaid with a 2:1 mixture of growth medium and microcrystalline cellulose (3.6% Avicel, RC-591, DuPont Pharma, Wilmington, DE, USA) including TPCK-treated trypsin (0.5 µg mL$^{-1}$) and incubated at 37 °C. At 3 days post infection, MDCK cells were fixed with 10% buffered formalin for 1 h. After removing the overlay, MDCK cells were permeabilized with PBS containing 0.5% Triton X-100 and 20 mM glycine for 20 min. The plates were washed 3 times with PBS containing 0.05% Tween 20 (PBST), and plaques were immunostained with a 1:3000 dilution of anti-IAV-NP antibodies (MAB8257 and MAB8258, 1:1 mixture, EMD Millipore, Saint Louis, MO, USA) and peroxidase-labeled goat anti-mouse IgG (SeraCare Life Sciences, Milford, MA, USA). Immunostained plaques were treated with peroxidase substrate (TrueBlue, SeraCare) according to the manufacturer's protocol and counted manually using an inverted light microscope (Laxco, Mill Creek, WA, USA). SARS-CoV-2 titers were determined by standard plaque assay on SV-VeroE6 cells.

*4.2. Antiviral Assessment*

Bacdown QAC (D + E) and test articles (A, B, C) were serially diluted 1:5 in DPBS from 1:5 out to 1:3125 (Figure 3). Virus stocks were thawed and diluted to $5.0 \times 10^5$ FFU/mL (H1N1) or PFU/mL (MOBAL-05) in DPBS. An aliquot of diluted virus was mixed with an equal volume of each QAC or test article dilution, or an equal volume of DPBS as a control, and incubated at room temperature for 1 h. To assess cytotoxicity of the compounds, each QAC or test article dilution was mixed with an equal volume of DPBS and incubated at room temperature for 1 h.

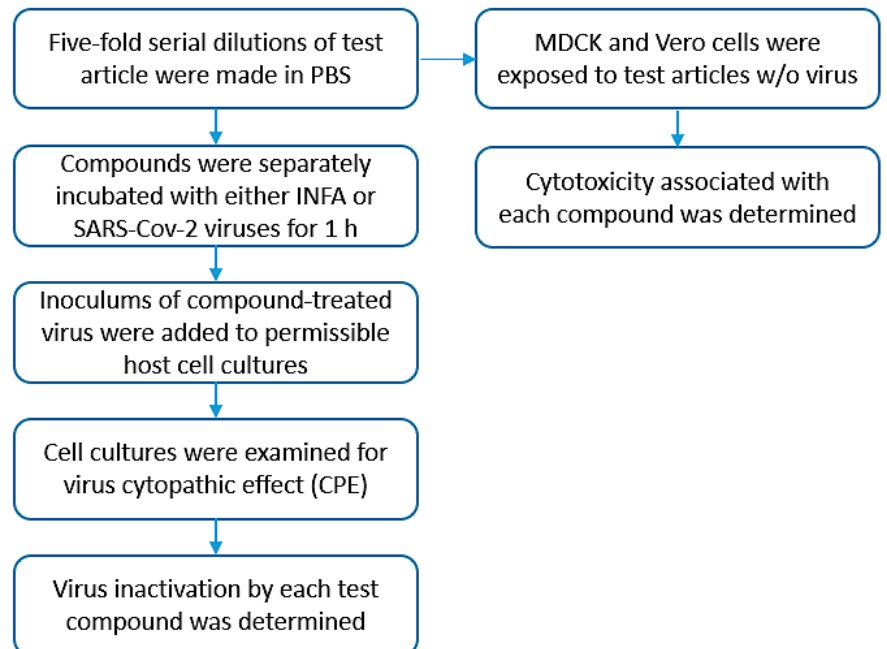

**Figure 3.** Testing schematic. Simplified diagram outlining general steps accomplished during the testing process.

After incubation, the 50% tissue culture infectious dose ($TCID_{50}$) was assessed. For H1N1, MDCK cells were plated in 96-well tissue culture plates (Alkali Scientific, Ft. Lauderdale, FL, USA) at a density of $1.25 \times 10^4$ cells/well in serum-free media and incubated at 37 °C with 5% $CO_2$ for 18–24 h to reach a confluency of ~60–70%. Samples were added to wells of 96-well dilution plates (CELLTREAT, Pepperell, MA, USA) and serially diluted 1:10 in serum-free media containing 0.5 µg/mL TPCK-trypsin from 1:10 out to 1:10,000. Media was removed from cells and replaced with 0.1 mL of sample dilutions. Cells were then incubated at 37 °C with 5% $CO_2$ and monitored daily for CPE. Final CPE scores were recorded at 48 h post-infection. $TCID_{50}$ titers were calculated according to the methods of Reed and Muench [16]. The limit of detection (LOD) for this assay was 55 $TCID_{50}$/mL.

For $TCID_{50}$ assessment of MOBAL-05, SV-VeroE6 cells were plated in 96-well tissue culture plates at a density of $2.0 \times 10^4$ cells/well in complete media and incubated at 37 °C with 5% $CO_2$ for 18–24 h to reach a confluency of ~60–70%. After incubation, samples were added to wells of 96-well dilution plates and serially diluted 1:10 in maintenance media from 1:10 out to 1:10,000. Media was removed from cells and replaced with 0.1 mL of sample dilutions, and cells were incubated at 37 °C with 5% $CO_2$ and monitored daily for CPE. Final CPE scores were recorded at 72 h post-infection.

*4.3. Cytotoxicity Assessment*

Cell cytotoxicity associated with each test article was also assessed in MDCK cells and SV-VeroE6 cells by exposing cells in 96-well plates to dilutions of each test article at each concentration and observing cells under the microscope for cell death after 48 and 72 h incubation, respectively. The 50% tissue culture cytotoxic dose ($TCCD_{50}$) of each compound

at each concentration was calculated using the method of Reed and Muench as above. The LOD for this assay was 55 $TCCD_{50}$/mL.

## 5. Conclusions

Initial testing of enveloped RNA viruses INFA and SARS-CoV-2 accomplished in the current study suggests that the boronium salts may be effective in controlling and/or eliminating enveloped viruses. Additionally, the benzylated salt form (B) demonstrated decreased cell toxicity, when compared to Bacdown QAC (D+E). If true, boronium salts would represent a viable option to using QAC disinfectants, which have known health and environmental consequences. Given the positive results from additional studies, a defined mixture of both boronium salts (A, B) containing varying alkyl tail lengths may result in an extremely effective disinfectant formulation that controls a broad spectrum of bacterial, fungal, and viral pathogens, the outcome being a new class of boronium ion-based disinfectants with fewer health hazards and environmental concerns than QACs.

**Author Contributions:** Conceptualization, J.H.D.J., T.J.R. and M.S.; Methodology, T.J.R., J.O.R. and R.W.R.; Validation, J.O.R. and R.W.R.; Formal Analysis, T.J.R. and J.O.R.; Investigation, T.J.R., J.O.R. and R.W.R.; Resources, J.H.D.J.; Writing—Original Draft Preparation, T.J.R. and J.O.R.; Writing—Review & Editing, J.H.D.J., T.J.R., J.O.R., R.W.R. and M.S.; Visualization, T.J.R. and J.O.R.; Supervision, T.J.R. and J.O.R.; Project Administration, T.J.R. and J.O.R.; Funding Acquisition, J.H.D.J. All authors have read and agreed to the published version of the manuscript.

**Funding:** This research was funded in part (synthesis and characterization of boronium salts) by National Science Foundation, grant number CHE-2102978.

**Institutional Review Board Statement:** Not applicable.

**Informed Consent Statement:** No human or animal subjects were used in this study.

**Data Availability Statement:** Available upon request.

**Acknowledgments:** We thank the National Science Foundation for supporting this work.

**Conflicts of Interest:** The authors declare no conflict of interest.

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
