# Peer review of "Boronium Salt as an Antiviral Agent against Enveloped Viruses Influenza A and SARS-CoV-2"

_2813-0464, doi:10.3390/applbiosci1030018_

Round 1
Reviewer 1 Report
Overall, the science is pretty straightforward. The data is easy to interpretate and supportive to the conclusion. However, it's will be better if there was more experimental data to support the mechanism part. There are some minor adjustment might need to be done as following
1.For Fig1., it's easier to read is the label can be placed on top of the figure.
2. While referring differences in antifungal or antibacterial activity, It’s better to add common why it showed the difference. For example when the author mention reference 4 and 6, mentioning some conclusion of the references would be better.
3. line 171 “The QAC benzalkonium”, b is bold, please correct it
4. line 231~248, it is more like the content should be in the introduction.
5. More comments can be found in the attachment.

Author Response
1. For Fig1., it's easier to read is the label can be placed on top of the figure.
Figure caption moved, as was suggested.
2. While referring differences in antifungal or antibacterial activity, It’s better to add common why it showed the difference. For example when the author mention reference 4 and 6, mentioning some conclusion of the references would be better.
A new paragraph has been added to the Introduction section to address this item.
3. line 171 “The QAC benzalkonium”, b is bold, please correct it
Bold font was not seen on the copy available for download, but will be examined for on the final copy.
4. line 231~248, it is more like the content should be in the introduction.
This information has been moved to the Introduction section.
Additional Comments:
4. Most of the explanation of the results especially the mechanism part seems with a lot of assumption. It would be better if the author can design the experiment or refer results from other articles with more experiments to support the assumption.
We appreciate the reviewer’s observations. Additional information about computational modeling has been added to the Discussion section. It offers support to the noted differences in each boronium salt’s ability to disrupt viral envelope phospholipids, leading to their inactivation. This represents a similar virus inactivation mechanism attributed to QACs on Influenza A and SARS-CoV-2 envelopes. We ask the reviewer to also keep in mind that these boronium salts are unique compounds that were recently developed and had not been previously studied for microbe control. As such, there is a relative lack of literature to draw further support from.
5. The introduction seems a little disconnected with the objective of the paper, which is the concentration breakpoints. It would be more connected If author could add more references about the different mechanism and how concentration breakpoints will be affected by it.
A paragraph has been added to the Introduction section to better explain the study rationale and provide a more even transition to the last paragraph.
Reviewer 2 Report
The manuscript is based on research in the scientific literature and presents information that the boronium salts may be effective in controlling and/or eliminating enveloped viruses. Boronium salts would represent a viable option to using QAC disinfectants having known health and environmental consequences.
My opinion is that maybe it would have been good for the authors to do studies on normal human lines as well.
Why were Madin-Darby canine kidney cells (MDCK cells) maintained in culture without fetal serum?
In general, the quality of the article is good and, overall, the manuscript is interesting to readers. English language and style are good, but there are some minor spelling mistakes. In conclusion, I consider the article could be a useful contribution to the journal. I recommend the manuscript for being published.
Author Response
My opinion is that maybe it would have been good for the authors to do studies on normal human lines as well.
Your opinion is noted. The intent is to determine if the compounds inactivate the virus so it's arguable that using the most permissive cells would provide the most rigorous assessment of this question. Human cells would be used in follow on studies to further assess the toxicity of these compounds.
Why were Madin-Darby canine kidney cells (MDCK cells) maintained in culture without fetal serum?
The authors can offer three reasons as to why fetal serum was not used. First, influenza infection requires activation with trypsin which is inhibited by serum, so best to just use serum free medium for all steps. Second, the master cell stocks are intended to be used for production of influenza vaccines, so to prevent additional purification steps serum free medium is used. Third, serum is expensive and may be a source for contamination, so again it is best to use serum free medium.
Reviewer 3 Report
Boronium Salt Demonstrates Potential as an Antiviral Agent Against Enveloped Viruses Influ-2 enza A and SARS-CoV-2
· Title: Change it to: Boronium Salt as an Antiviral Agent Against Enveloped Viruses Influ-2 enza A and SARS-CoV-2
· Abstract: Recommendation sentence is missing at the end of abstract. Write it as last sentence
· Put space between number and degree symbol. For degree symbol, insert it from symbol. and number.
· L285: After 72 h 0.2 ml : Add comma after h
· The study is good; well written but statistical analysis was not seen. All scientific data have to be statistically analyzed. It was better to do experiments in triplicate in order to get tangible results. If they still have samples, they can repeat last two steps.
· The main question was to investigate if the two boronium salt derivatives can inactivate influenza A and 14 SARS-CoV-2 viruses. It is not well explained in the introduction. Problem statement no clear in the last paragraph of introduction. Even some sentences are incomplete. This paragraph has to be revised.
· Full stop have to be removed from caption of table 1 and 2
Author Response
Title: Change it to: Boronium Salt as an Antiviral Agent Against Enveloped Viruses Influ-2 enza and SARS-CoV-2
Recommended change was accomplished.
Abstract: Recommendation sentence is missing at the end of abstract. Write it as last sentence
Recommendation added as last sentence of abstract.
Put space between number and degree symbol. For degree symbol, insert it from symbol. and number.
Corrected.
L285: After 72 h 0.2 ml : Add comma after h
Corrected.
The study is good; well written but statistical analysis was not seen. All scientific data have to be statistically analyzed. It was better to do experiments in triplicate in order to get tangible results. If they still have samples, they can repeat last two steps.
We appreciate the reviewer's comments. The objective of the current investigation was to establish starting sample concentrations for further investigation and not to definitively establish a difference between test article antiviral activity. Given limited resources, a single determination was considered sufficient in accomplishing this aim. However, replicate samples will be used in all future studies to determine whether or not results are statistically significant.
The main question was to investigate if the two boronium salt derivatives can inactivate influenza A and 14 SARS-CoV-2 viruses. It is not well explained in the introduction. Problem statement no clear in the last paragraph of introduction. Even some sentences are incomplete. This paragraph has to be revised.
An additional paragraph has been added to the Introduction section to fill in this gap in information. The authors would like to thank the reviewer for making us aware of this issue.
Full stop have to be removed from caption of table 1 and 2
Did not detect full stops in table captions, but will correct if seen on final copy.
Round 2
Reviewer 1 Report
Authors have addressed all my questions and did the modification I was asking. I think the current form is good for publishing.
Reviewer 3 Report
The comments were addressed as suggested. However, the text from L51 to L61 has to be written in paragraph, not in point form. They have to be separated by " ; " not (.)